# A Linearly Polarized Wavelength-Tunable Q-Switched Fiber Laser with a Narrow Spectral Bandwidth of 112 MHz

**DOI:** 10.3390/s23115128

**Published:** 2023-05-27

**Authors:** Chengjie Zhu, Xuezong Yang, Yuxuan Liu, Muye Li, Yuxiang Sun, Wei You, Peng Dong, Dijun Chen, Yan Feng, Weibiao Chen

**Affiliations:** 1Shanghai Institute of Optics and Fine Mechanics, Chinese Academy of Sciences, Shanghai 201800, China; 2School of Physics and Optoelectronic Engineering, Hangzhou Institute for Advanced Study, University of Chinese Academy of Sciences, Hangzhou 310024, Chinayx.sun.xy@hotmail.com (Y.S.); 3School of Fundamental Physics and Mathematical Sciences, Key Laboratory of Gravitational Wave Precision Measurement of Zhejiang Province, Taiji Laboratory for Gravitational Wave Universe, Hangzhou Institute for Advanced Study, University of Chinese Academy of Sciences, Hangzhou 310024, China

**Keywords:** fiber laser, Q-switched fiber laser, saturable dynamic induction grating, narrow bandwidth, tunable wavelength

## Abstract

A tunable and narrow-bandwidth Q-switched ytterbium-doped fiber (YDF) laser is investigated in this paper. The non-pumped YDF acts as a saturable absorber and, together with a Sagnac loop mirror, provides a dynamic spectral-filtering grating to achieve a narrow-linewidth Q-switched output. By adjusting an etalon-based tunable fiber filter, a tunable wavelength from 1027 nm to 1033 nm is obtained. When the pump power is 1.75 W, the Q-switched laser pulses with a pulse energy of 10.45 nJ, and a repetition frequency of 11.98 kHz and spectral linewidth of 112 MHz are obtained. This work paves the way for the generation narrow-linewidth Q-switched lasers with tunable wavelengths in conventional ytterbium, erbium, and thulium fiber bands to address critical applications such as coherent detection, biomedicine, and nonlinear frequency conversion.

## 1. Introduction

Due to their characteristics of a short pulse duration and high peak power, Q-switched lasers are widely used in optical communication, detection, and sensing [1,2,3,4] and have been extensively investigated in recent years [5,6,7]. In some specific applications, such as nonlinear frequency conversion, biomedicine, and coherent detection, short-pulse lasers with narrow bandwidths have an excellent performance and attractiveness [8,9,10,11]. However, laser operation with both a narrow bandwidth and a short pulse is challenging. Q-switching and bandwidth-limiting elements are the keys to achieving pulsed emission and spectral narrowing. Saturable absorbers and Bragg gratings are commonly used as passive Q-switching and spectral-filtering components, respectively. For example, Hana Ouslimani et al. reported a narrow-linewidth Q-switched DBR laser operating at 1030 nm with a pulse width of 12 ns, linewidth of 372 MHz, and repetition rate of 21 kHz, which was obtained through the combination of a fiber Bragg grating (FBG) spectral filter and a dye-doped polymer film saturable absorber [8]. This method involved both the optical fiber and spatial optics which determine the requirements for the complicated coupling technique. In comparison, all-fiber structures tend to have a more optimized design and stability. Usha Chakravarty et al. employed a narrow-linewidth and tunable Q-switched YDF laser using an acousto-optic modulator and multimode interference filter in a linear bulk resonator and an all-fiber ring resonator, respectively [9]. Both resonators achieved tuning ranges above 12 nm and spectral linewidths below 0.1 nm. Although the linear resonator with a combination of spatial and fiber structures had an average output power of 500 mW, higher than the power of 10 mW for the all-fiber ring resonator, the all-fiber resonator clearly reduced the intricacy of the total devices and could potentially be used for further power amplification. Yao Bo et al. continued to study the all-fiber-structured narrow-linewidth Q-switched laser [10] using a specially designed low-reflectivity cladding power stripper, a narrowband FBG, and a length-optimized ytterbium-doped single-cladding fiber, which were able to efficiently suppress the ASE gain self-saturation and quickly establish the Q-switched pulses. Additionally, a watt-level Q-switched output with a bandwidth of 0.15 nm, pulse width of 9 ns, and repetition rate of 175 kHz was achieved. However, the above-mentioned works used separate components to achieve spectral narrowing and saturable absorption functions, which lead to a highly complex cavity design and limited cost savings. Recently, the all-fiber structure using an unpumped rare-earth-doped active fiber as the saturable absorber, as well as a dynamic Bragg grating acting as a spectral filter, has attracted increasing attention as a method to achieve narrow-linewidth continuous-wave (CW) or Q-switched lasers [11,12,13]. Zhang Junxiang et al. proposed a single-longitudinal-mode thulium fiber laser with a spectral linewidth of 4.4 kHz and output power of 407 mW at 1720 nm utilizing an unpumped thulium-doped fiber (TDF) for providing saturable absorption and linewidth compression [12]. Furthermore, Wen Zengrun et al. inserted a section of unpumped erbium-doped fiber (EDF) between a circulator and an FBG centered at 1550 ± 0.25 nm to form a saturable dynamic induction grating (SDIG) in order to accomplish narrow-bandwidth Q-switched laser operation [11]. To obtain standing-wave-intensity refractive modulation, which is the core SDIG technique, an FBG was utilized as the reflective mirror, resulting in a fixed monotonic output wavelength. The narrowest spectral linewidth of 29.1 pm was achieved when the SDIG length and pump power were 20 cm and 250 mW, respectively. The technical scheme and the main output characteristics of the above-mentioned narrow-linewidth Q-switched lasers, including the output wavelength, output power, and so on, are shown in Table 1.

In this paper, a wavelength-tunable and narrow-bandwidth Q-switched ytterbium-doped fiber (YDF) laser was investigated using an all-polarization-maintained (PM) fiber ring cavity. A section of unpumped YDF was inserted between a Sagnac loop mirror and the circulator, with a dynamic spectral-filtering grating. An etalon-based wavelength selector and a broadband Sagnac loop reflector supported the laser wavelength tuning. At the pump power of 1.75 W, the Q-switched laser pulsed with a pulse energy of 10.45 nJ, repetition frequency of 11.98 kHz, and spectral linewidth of 112 MHz. The wavelength was continuously adjusted from 1027 nm to 1033 nm.

## 2. Experimental Setup

Figure 1 shows the experimental setup of the wavelength-tunable narrow-linewidth Q-switched laser. The pump source is a laser diode (LD) of 976 nm with a maximum output power of 27.1 W and a 3 m long double-cladding PM YDF (Coherent PLMA-YDF-10/125-M) which has a cladding absorption of 4.95 dB/m at 976 nm and is used as the gain fiber. A PM cladding power stripper (CPS) is placed behind the gain fiber to remove the residual pump light. A circulator is used to ensure the clockwise operation of the light in the cavity while introducing Q-switching, band-limiting, and wavelength-tuning elements into the cavity. The same type of 1 m long PM YDF is inserted behind the circulator’s port 2, and a PM fiber-tunable filter (TF) with a wavelength tunability from 1020 nm to 1095 nm is placed behind YDF2. The output ports of the PM coupler 1, with a split ratio of 49/51 at 1064 nm, are fused together to form a Sagnac loop mirror, which provides a high reflectance from 1030 nm to 1080 nm. The devices placed after the circulator’s port 2 provide various modulation effects in the laser resonator. The unpumped YDF2 (absorption loss of approximately 1.5 dB/m at 1030 nm) provides the saturable absorption effect to boot the Q-switching. The forward- and backward-propagating lights travelling through YDF2 form a standing-wave dynamic refractive grating which together with the Sagnac loop mirror and the TF construct a tunable spectral filter to achieve wavelength tunability and a narrow-linewidth output. Finally, the Q-switching pulses are exported through the 20% port of the optical coupler (coupler 2). An integrating sphere photodiode power sensor (Thorlabs, S145C) and an InGaAs Biased Detector (Thorlabs, DET08CFC/M) are used to measure the output power and detect the pulse profiles.

## 3. Experimental Results

In the experiment, the lengths of YDF1 and YDF2 were 3 m and 1 m, respectively. The TF transmission spectrum was first tuned to 1030 nm, at which the unpumped fiber of YDF2 had a more appropriate saturable absorption than that at longer wavelengths. The Q-switched pulse reached the threshold when the pump power was increased to 1.75 W, and after the start of the pulsed operation, the stable Q-switching was maintained in the pump power range of 1.3 W to 1.75 W. The appearance of the power bistability was mainly related to the saturable absorber properties of the unpumped YDF2 before the laser oscillation [12]. After reaching the lasing threshold, the absorption loss of the unpumped YDF2 decreased significantly due to the high laser intensity in the cavity. Therefore, the reduction in the pump power within a certain range was unable to hinder the laser oscillation. The output average power (blue) and single-pulse energy (red), as a function of the pump power, are shown in Figure 2. At the maximal pump power of 1.75 W, the output power and pulse energy reached the highest values of 3.37 mW and 10.45 nJ, respectively.

Figure 3a shows the Q-switched pulse train recorded using a digital oscilloscope (Agilent Technologies, Santa Clara, CA, USA, DSOX2024A) with a bandwidth of 200 MHz. At the pump power of 1.75 W, the pulse repetition frequency was 11.54 kHz. Between the main pulses, there were four small pulses with a repetition rate of approximately 59.08 kHz, which were induced by relaxation oscillation in a saturable absorption Q-switch period. The Q-switched pulse duration is shown in Figure 3b, and the full width at half maximum (FWHM) was 2.96 μs. By varying the pump power, the output pulse width and pulse repetition frequency were measured, and they are shown in Figure 3c and Figure 3d, respectively. For the quantities in Figure 3c, at each pump power, five pulses were measured, and the duration was the average value. When the pump power was tuned from 1.3 W to 1.75 W, the FWHM of the pulse duration was basically unchanged between 3.41 µs and 3.09 µs, and the repetition frequency of the pulse train increased from 5.61 kHz to 11.54 kHz. This linear increase in the pulse repetition rate versus pump power was due to the shortening of the saturable absorption time at a high intracavity intensity. Similarly, the repetition frequency of the relaxation oscillation pulses increased with the increase in pump power.

When acting as a Q-switching modulator, the unpumped YDF2, together with the fiber Sagnac loop mirror, also functioned to provide a spectral-narrowing effect. The optical wave ejected from the circulator port 2 and the backward optical wave reflected by the Sagnac loop mirror formed a standing-wave field, which induced the SDIG through the unpumped YDF2. The SDIG can be considered as a Bragg reflective grating that functions to provide the spectral-narrowing effect [14]. The wavelength-related spectral transmission bandwidth of the SDIG is calculated as follows [15]:(1)$$∆f=\frac{c}{\lambda }\kappa \sqrt{{\left(\frac{∆n}{2n_{\mathrm{eff}}}\right)}^{2}+{\left(\frac{\lambda }{2n_{\mathrm{eff}}L_{g}}\right)}^{2}},$$
where c is the speed of light in a vacuum. ∆n is the variation in the refraction index of the fiber. $n_{\mathrm{eff}}$ and $L_{g}$ are the refractive index and the length of YDF2, respectively. $\kappa =2∆n/{(\lambda n}_{\mathrm{eff}})$ is the coupling coefficient of the SDIG. In the experiment, a Fabry–Perot scanning interferometer (FPI, Thorlabs, SA210-8B) with a free spectral range (FSR) of 10 GHz and spectral resolution of 67 MHz was used to measure the linewidth of the output laser. Figure 4a shows the FPI transmission spectrum and the zoomed-in single spectral peak at the pump power of 1.75 W. The sweeping period of the FPI was set to 6.3 ms. The frequency interval between the two main peaks was the FPI FSR of 10 GHz, and the FWHM of a single peak corresponded to the spectral linewidth of 112 MHz. The theoretical spectral transmission bandwidth ∆f of the SDIG was calculated to be approximately 120 MHz, with a refraction variation ∆n of 7.12 × 10^−5^. Figure 4b shows the experimental laser linewidth as a function of the pump power, and the red dashed line is the fitted curve. It shows that the laser spectral linewidth gradually narrowed from 310 MHz to 120 MHz with the increase in pump power and tended to stabilize to around 130 MHz when the pump power was higher than 1.4 W.

The Sagnac fiber loop mirror was created by splicing together the two output ports of a fused-tapered fiber coupler. When the optical intensity through the fiber loop mirror is moderate, the nonlinear reflectivity of the Sagnac loop mirror can be ignored [16]. The linear reflectivity R of a Sagnac loop mirror is described by [17]:R = 4α(1 − α)L(λ),(2)
where α represents the coupling ratio of the two output terminals, L(λ) represents the loss term of the Sagnac loop mirror at the wavelength of λ, and the test value of L(λ) is 0.96 at 1064 nm. The fiber coupler 1 used in our experiment had a splitting ratio of 49:51 at 1064 nm. Substituting the values into Equation (2), the reflectivity of the Sagnac loop mirror at 1064 nm was calculated as 95.96%. Using a circulator and considering the insertion loss, the reflectivity of the Sagnac loop mirror was measured experimentally. The measured reflectivity at 1060 nm was 95.2%, which agreed well with the calculation result of 95.96%. Due to the broad working band of the fused-tapered coupler, the reflectance of the fiber loop mirror at wavelengths from 1030 nm to 1080 nm was recorded, and it is shown in Figure 5. The minimum reflectivity was 93.4% at 1030 nm and the maximum reflectivity was 96.8% at 1080 nm.

By continuously tuning the center wavelength of the TF, a narrow-linewidth Q-switched pulsed laser operating from 1027 nm to 1033 nm was achieved at the fixed pump power of 1.55 W. Figure 6a shows the intensity-normalized laser spectrum from 1027.22 nm to 1033.07 nm at approximately 1 nm intervals, which was recorded using an optical spectrum analyzer (AQ6370, Yokagawa) with a wavelength resolution of approximately 0.02 nm. The wavelength-tuning range was dependent on the characteristics of the two pieces of the YDF. When the working wavelength of the filter was tuned to be shorter than 1027 nm, the amplified spontaneous emission at 1030 nm in YDF1 increased sharply. Additionally, when the working wavelength of the filter was tuned to be longer than 1033 nm, the laser intensity greatly increased, and the unpumped YDF2 was bleached. The average output power and pulse repetition frequency, respectively, as a function of the laser wavelength, are shown in Figure 6b. When the laser wavelength was tuned to be from 1027.22 nm to 1033.07 nm, the output power increased from 1.5 mW to 2.9 mW and the pulse repetition frequency increased from 7.15 kHz to 10.64 kHz. This is why the gain in the YDF at 1033 nm was slightly higher than that at 1027 nm and the absorption loss at 1033 nm was less than that at 1027 nm. Figure 6c,d depicts the pulse duration and spectral linewidth, respectively, as a function of the lasing wavelength at the pump power of 1.55 W, where the pulse duration is relatively stable between 2.24 μs and 2.91 μs and the spectral linewidth fluctuates slightly between 61.62 MHz and 111.73 MHz. The minimum pulse width was 2.24 μs at 1031 nm, and the spectral linewidth reached the minimum of 61.62 MHz at 1027 nm.

## 4. Conclusions

In summary, an all-fiber-based tunable and narrow-bandwidth Q-switched YDF laser was investigated by employing a piece of unpumped YDF to form the SDIG for providing saturable absorption and the spectral filter simultaneously. At the maximal pump power of 1.75 W, Q-switched pulses with a pulse energy of 10.45 nJ, repetition frequency of 11.54 kHz, spectral linewidth of 112 MHz, and pulse duration of 2.97 μs were achieved. Using a fiber TF as the wavelength selector, the narrow-linewidth Q-switching laser was tuned from 1027 nm to 1033 nm at the pump power of 1.55 W. In this work, the unpumped YDF was crucial for the performance of the laser pulse characteristics. Optimizing the unpumped YDF fiber length or using a fiber with a higher absorption efficiency could improve the wavelength-tuning range and the output pulse energy. This work provides a compact, stable, and low-cost wavelength-tunable and narrow-bandwidth Q-switched laser, and this method is also applicable to other doped fibers, such as erbium and thulium fibers, for expanding the lasing band to 1.5 μm and 2 μm.

## Figures and Tables

**Figure 1 sensors-23-05128-f001:**
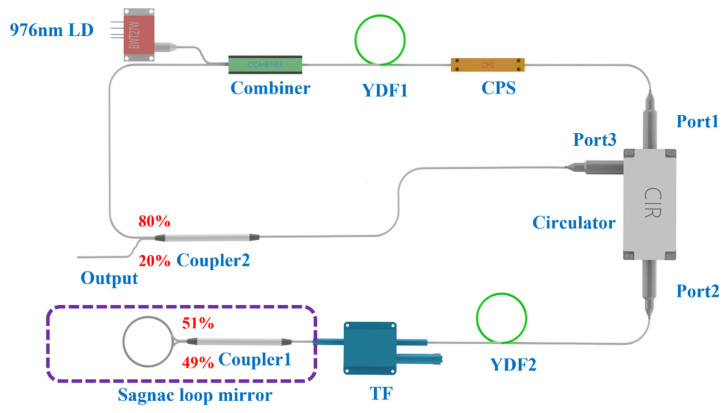
Experimental setup of the tunable and narrow-bandwidth Q-switched YDF laser. LD: laser diode, YDF: ytterbium-doped fiber, CPS: cladding power stripper, TF: tunable filter.

**Figure 2 sensors-23-05128-f002:**
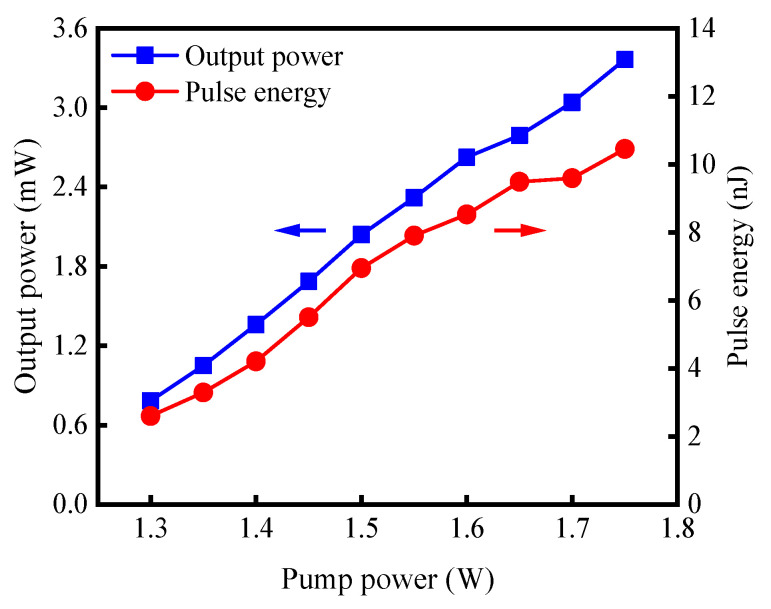
Output power (blue) and pulse energy (red) versus pump power.

**Figure 3 sensors-23-05128-f003:**
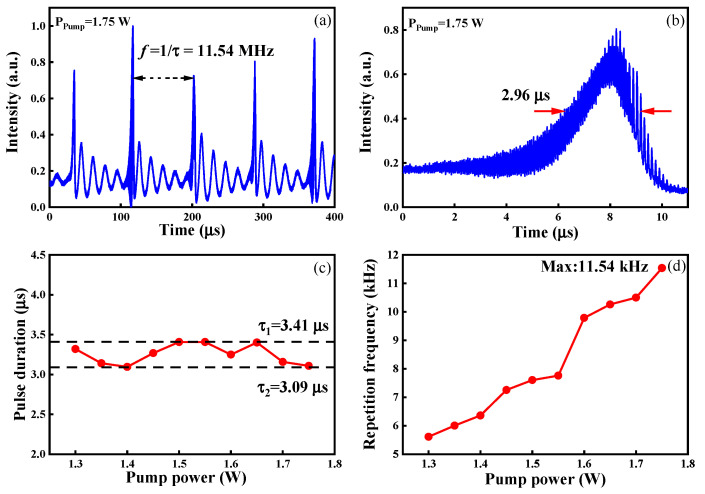
(**a**) Pulse train and (**b**) single-pulse profile at the pump power of 1.75 W. (**c**) Pulse duration and (**d**) pulse repetition frequency as a function of pump power.

**Figure 4 sensors-23-05128-f004:**
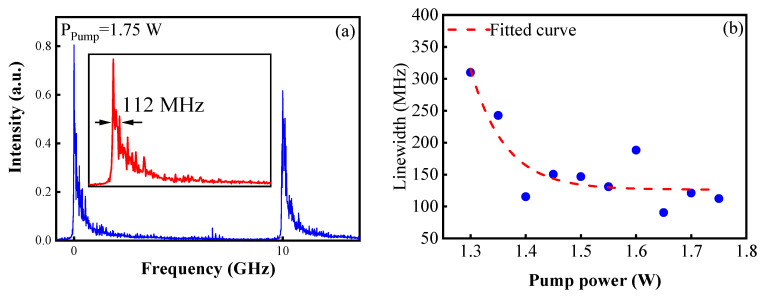
(**a**) FPI traces of the Q-switched pulses and a single main peak (inset) at a 1.75 W pump power. (**b**) Spectral linewidth (blue dot) versus pump power in the experiment and the fitted trend curve (red dashed).

**Figure 5 sensors-23-05128-f005:**
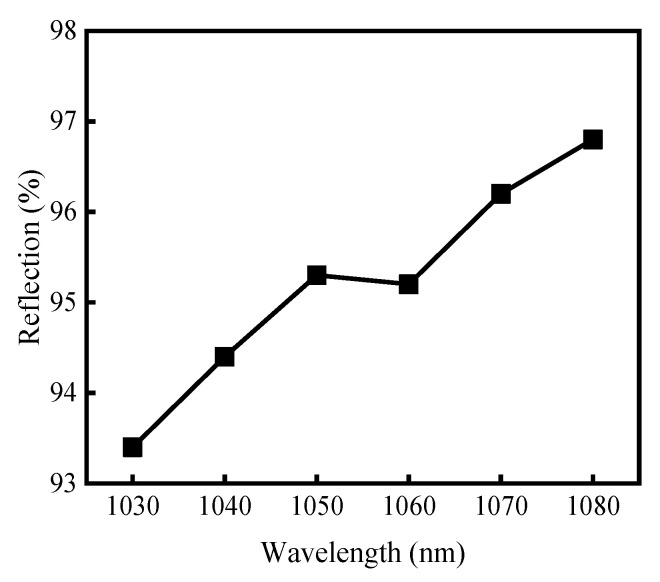
Reflectivity of the Sagnac loop mirror versus signal wavelength.

**Figure 6 sensors-23-05128-f006:**
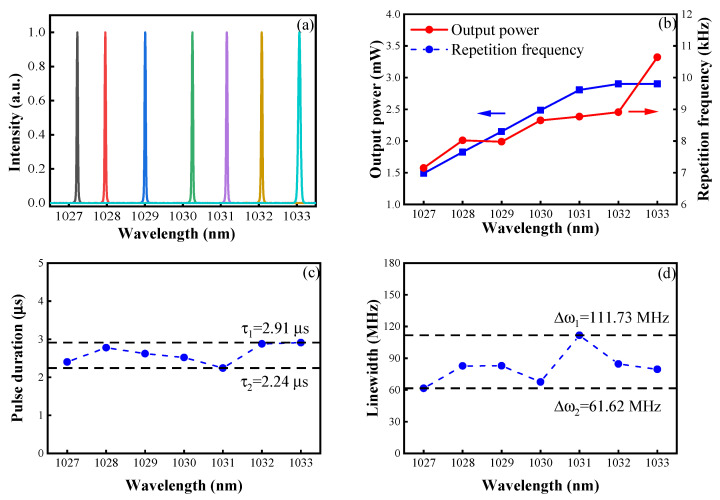
At the pump power of 1.55 W: (**a**) the normalized spectra from 1027 nm to 1033 nm, (**b**) output power and repetition frequency versus the wavelength, and (**c**) pulse duration and (**d**) spectral linewidth versus the wavelength, respectively.

**Table 1 sensors-23-05128-t001:** Performance of the narrow-linewidth Q-switched lasers.

Paper	Q-Switching Element	Filter Element	Wavelength	Power	Pulse Width	RF *	Linewidth
H. Ouslimani et al. [8]	A dye-doped polymer film saturable absorber	A distributed Bragg grating cavity	1030 nm	0.75 mW	12 ns	21 kHz	1.32 pm
U. Chakravarty et al. [9]	An acousto-optic modulator	A multimode interference filter	1038~1050 nm	100 mW	200 ns	20 kHz	<0.1 nm
B. Yao et al. [10]	An acousto-optic modulator	A single-mode FBG	1064 nm	1.3 W	9 ns	175 kHz	0.15 nm
Z. Wen et al. [11]	A saturable dynamic induction grating formed by inserting a segment of unpumped		1550 nm	250 mW	1.75 μs	60.6 kHz	29.1 pm
	EDF between a circulator and an FBG						

* RF: repetition frequency.

## Data Availability

The data that support the findings of this study are available from the corresponding author upon reasonable request.

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
