# Peer review of "A Linearly Polarized Wavelength-Tunable Q-Switched Fiber Laser with a Narrow Spectral Bandwidth of 112 MHz"

_sensors, 2023, doi:10.3390/s23115128_

Round 1

Reviewer 1 Report

In this work, Zhu and his colleagues conducted an experimental study to develop a tunable Q-switched ytterbium-doped fiber (YDF) laser with a narrow bandwidth. The authors utilized a non-pumped YDF as a saturable absorber and constructed a Sagnc interferometer to achieve spectral narrowing and wavelength tunability. The setup allowed them to create a saturable dynamic induction grating, which facilitated the operation of a narrow bandwidth Q-switched laser.

Upon careful review of the manuscript, I found the work to be interesting and of practical importance. The presentation is well-organized and written, enabling easy access to the reported findings without significant technical challenges. However, I have the following reservations that the authors should consider in their revision before I can recommend it for publication in Sensors:

(1) Justification of the importance and significance of the reported work: While the authors have mentioned some related works in the introduction, it would be beneficial for them to expand this discussion by comparing their results with the achievements documented in the literature. Fiber-based Q-switched lasers have been extensively studied, and there is a substantial body of research in this field. It would be constructive for the authors to create a table comparing their work to the milestone studies in this particular area. Merely reporting their own achievements might not be convincing enough.

(2) Results of the non-pumped YDF as a saturable absorber: Although the authors employed a non-pumped YDF as a saturable absorber, it would be more meaningful if they could provide experimental data to support this claim. Additionally, if this YDF exhibits absorption saturation, the effect should manifest in the pulse duration and repetition rate through nonlinear behavior. However, the reported data does not demonstrate such a trend. Could I be missing something here?

(3) Other issues: (a) There are a few grammar issues in the manuscript. (b) In Section 2, the authors state, "The pump source is a laser diode (LD) at 976 nm with a maximum output power of 27 W." Is this 27 W value accurate? (c) Regarding Figs. 2(b) and (c), how did the authors measure these quantities? As evident from the Q-switched pulse train depicted in Fig. 2(a), even the main pulses are not identical, some with higher amplitudes and others with relatively smaller peaks. Please provide a more detailed explanation of the measurement and data processing. (d) In line 155, the "red dotted line" should be referred to as the "red dashed line." (e) In line 160, "1.7 W" should be corrected to "1.75 W." (f) In line 162, "red dash" should be changed to "red dashed." (g) In line 197, how did the authors normalize the spectra for wavelengths ranging from 1027 nm to 1033 nm? (h) The authors mentioned that the maximal pump power is 1.75 W, which results in the maximal output power and pulse energy (see Figure 2). However, Figure 2 shows a gradual increase in output power and pulse energy as a function of pump power. I do not observe any reduction trend that might occur if the pump power exceeds 1.75 W. Please provide an explanation.

In summary, this work presents a technical improvement in demonstrating an all-fiber-based tunable and narrowband Q-switched YDF laser. The results are interesting and may have practical implications. However, before a final decision can be reached, I strongly urge the authors to thoroughly revise the manuscript by addressing all the issues outlined above.

The English is fine in general and the manuscript is well written and organized.

Reviewer 2 Report

In this paper, a wavelength-tunable and narrow bandwidth Q-switched ytterbium doped fiber laser is demonstrated by using an all-polarization-maintained (PM) fiber ring cavity. The manuscript is well structured, but there are still some points need to be achieved.

(1)The authors mentioned that the fiber tunable filter can provide a wavelength tunability from 1020 to 1095 nm. However, the experimental results showed that only an output wavelength from 1027 nm to 1033 nm was achieved. The authors should add some sentences to explain the reason.

(2)The average power and pulse energy of this Q-switched Yb-fiber laser were very low, only 3.4 mW and 10.5 nJ, respectively. Could the authors give any method to improve the output power? 

(3)Figure 1 is not clear enough and need to be improved.

Reviewer 3 Report

The authors introduced an all-fiber-based tunable and narrow bandwidth Q-switched YDF laser. The manuscript is well organized with sufficient experimental results. I think it can be accepted after a minor revision.   1. In the description of Figure 1, why a split ratio of 49/51 for Coupler1 is used?   2. The quantitative comparison of the performance between the proposed Q-switched laser to other all-fiber-based Q-switched lasers is highly suggested.   3. Since the tunable spectrum range of the proposed laser is relatively small (from 1027-1033 nm), is there any methods that can be used to expand the spectral range over 10nm?

-

Round 2

Reviewer 1 Report

In this revised version, the authors have responded to my previous comments and have made appropriate changes in the manuscript. Although the presentation could be further improved, I am happy with this revised one. Therefore, I recommend it for publication in Sensors.